

# The aerosol pathway is crucial for observationally constrained climate sensitivity and anthropogenic forcing

Ragnhild Bieltvedt Skeie[1], Magne Aldrin[2], Terje K. Berntsen[3], Marit Holden[2], Ragnar Bang Huseby[2], Gunnar Myhre[1], Trude Storelvmo[3]

[1]CICERO Center for International Climate Research, P.O. Box 1129 Blindern, 0318 Oslo, Norway

[2]Norwegian Computing Center, P.O. Box 114 Blindern, 0314 Oslo, Norway

[3]Department of Geosciences, University of Oslo, P.O. Box 1047 Blindern, 0316 Oslo, Norway

*Correspondence to*: Ragnhild Bieltvedt Skeie (r.b.skeie@cicero.oslo.no)

**Abstract.** Climate sensitivity and aerosol forcing are two of the most central, but uncertain, quantities in climate science that are crucial for assessing historical climate as well as future climate predictions. Here, we use a Bayesian approach to estimate the inferred climate sensitivity and aerosol forcing using observations of temperature and global ocean heat content and prior knowledge of effective radiative forcing (ERF) over the industrial period. Due to limited information on uncertainties related to the time evolution of aerosol forcing, we perform a range of sensitivity analyses with idealized aerosol time evolution. The estimates are sensitive to the aerosol forcing pathway with the mean estimate of inferred climate sensitivity ranging from 2.0 to 2.4 K, present-day (2019 relative to 1750) aerosol ERF ranging from -0.7 to -1.1 W m$^{-2}$ and anthropogenic ERF ranging from 2.6 to 3.1 W m$^{-2}$. Using observations and forcing up to and including 2022, the inferred effective climate sensitivity is 2.2 K with a 1.6 to 3.0 K 90% uncertainty range. Analysis with more freely evolving aerosol forcing between 1950 and 2014 shows a strong negative aerosol forcing trend in the latter part of the 20th century that is not consistent with observations. Although we test our estimation method with strongly idealized aerosol ERF pathways, our posteriori estimates of the climate sensitivities end up in the weaker end of the range assessed in the Sixth Assessment Report of the Intergovernmental Panel on Climate Change (IPCC AR6). As our method only includes climate feedbacks that have occurred over the historical period, it does not include the pattern effect, i.e. where climate feedbacks are dependent on the pattern of warming which will likely change into the future. Adding the best estimate of the pattern effect from IPCC AR6, our climate sensitivity estimate is almost identical to the IPCC AR6 best estimate and very likely range.

## 1 Introduction

Historically, anthropogenic aerosols have partly masked the greenhouse gas-driven warming due to their general cooling effect. The magnitude of this aerosol cooling over the past century is one of the main uncertainties in our understanding of



historical climate change (Forster et al., 2021) and a limiting factor for future climate projections (Watson-Parris and Smith, 2022).

Climate feedbacks, especially those governed by highly parameterized processes in climate models (e.g. cloud feedbacks), are another large source of uncertainty in climate predictions (Hawkins and Sutton, 2009) and a reason for the large spread in climate sensitivity in climate models (Zelinka et al., 2020;Sherwood et al., 2020). The total climate feedback strength is commonly quantified by the equilibrium climate sensitivity (ECS), defined as the equilibrium surface air temperature change following a doubling of atmospheric $CO_2$ concentration. Complementary to ECS quantified from climate model simulations, observed historical climate change can be used to constrain the total feedback strength. However, this method is limited by the uncertain historical forcing of the climate (Gregory et al., 2020;Forster, 2016;Knutti et al., 2017) where aerosol forcing is the main source of uncertainty (Forster et al., 2021;Forster et al., 2024). The forcing of the climate system is commonly expressed as effective radiative forcing (ERF), defined as the change in the Earth Energy Balance relative to pre-industrial conditions due to a change in an external driver of climate change and including adjustments to this forcing that are not mediated by changes in surface temperature (Sherwood et al., 2015). For aerosols, adjustment processes especially related to clouds are important and contribute to the large uncertainties in aerosol ERF (Bellouin et al., 2020). Importantly, both the magnitude of the present-day total aerosol ERF as well as the historical pathway of aerosol ERF – how aerosols have evolved over time – contribute to the uncertainty in observational constraints on climate sensitivity. Both the climate feedbacks and aerosol ERF are crucial for assessing historical climate change using models (Gillett et al., 2021).

For several models contributing to the 6[th] phase of the Coupled Model Intercomparison Project (CMIP6), modeled temperatures in the mid to late 20[th] century are colder than observed (Flynn and Mauritsen, 2020) and show a more rapid warming than in the observations since the early 1980s (Tokarska et al., 2020). Studies point to a too strong aerosol ERF as part of the reason for the mismatch between modeled and observed temperatures in the second half of the 20[th] century (Flynn et al., 2023;Flynn and Mauritsen, 2020;Gillett et al., 2021;Smith and Forster, 2021;Zhang et al., 2021). Aerosol ERF is diagnosed from models contributing to CMIP6 within Radiative Forcing Model Intercomparison Project (Pincus et al., 2016) (RFMIP) and the Aerosol Chemistry Model Intercomparison Projects (Collins et al., 2017) (AerChemMIP). It is noteworthy that the aerosol ERF time evolutions from these models, driven by the same emission inventory, show considerable variation with the timing of the peak in negative total aerosol ERF varying from 1975 to 2010 (Smith et al., 2021b).

Estimated climate sensitivity based on historical observations has also been found to be sensitive to different aerosol forcing pathways (Skeie et al., 2018). In IPCC AR6 time series of ERF from 1750 to 2019 were presented for a range of climate forcers (Forster et al., 2021) and more recently these have been extended to 2022 (Forster et al., 2023) and 2023 (Forster et al., 2024). The uncertainties in these forcing time series are presented as the 5[th] and 95[th] percentiles, but uncertainties in the time evolution of these forcings are not quantified. As shown in Smith et al. (2021b), the different representations of model physics result in different pathways of the total aerosol ERF. In addition, uncertainties in aerosol and aerosol-precursor emissions will add additional uncertainties to the aerosol time evolution (Smith et al., 2021b).





In the emission inventory provided for CMIP6 there are no quantifications of uncertainties (Hoesly et al., 2018). For
historical global anthropogenic $SO_2$ emissions, uncertainties of 8-14% (5 to 95 % confidence interval) have previously been
estimated, while regional emission uncertainties are larger (Smith et al., 2010). For black and organic carbon emissions from
fuel combustions, uncertainties in global emissions are larger than a factor of two (Bond et al., 2007). The lifetime of
aerosols is short, on the order of days (Samset et al., 2014;Textor et al., 2007) and both chemical conversion in the
atmosphere (Manktelow et al., 2007) as well as the forcing efficiency (Shindell et al., 2015;Kasoar et al., 2018) are
dependent on the location of emissions. Therefore, uncertainties in geographical distribution and their trends, as well as the
total amount of emissions, will add additional uncertainties to aerosol ERF as diagnosed in the models.

Due to air pollution quality controls, the global emissions of $SO_2$ have been rapidly decreasing, first due to emission controls
in Europe and North America (Hoesly et al., 2018;Aas et al., 2019) and over the latest decade in East Asia (Zheng et al.,
2018). From both observations and modelling efforts, there is now robust evidence of a reversal of the trend in aerosol ERF,
with reduced cooling contribution from aerosols over the last decades (Quaas et al., 2022).

Here, building on previous work (Skeie et al., 2018, 2014;Aldrin et al., 2012), we investigate the importance of aerosol ERF
time evolution for observational constraints on ERF and climate sensitivity. The estimated climate sensitivity is the Inferred
Effective Climate Sensitivity ($ECS_{inf}$) and for known reason $ECS_{inf}$ differs from ECS as calculated in climate models (e.g.
Armour et al., 2024;Sherwood et al., 2020) which we will discuss in the Discussion section. We use a Bayesian framework
considering an energy balance model and use the most up to date ERF time series and observations of temperature and ocean
heat content to constrain the ERF and $ECS_{inf}$. Using a range of idealized aerosol ERF evolutions, we assess the sensitivities
of these estimates to the aerosol pathway. Finally, we also allow the ERF for aerosol-cloud interaction to freely evolve
within the IPCC AR6 uncertainty range in the most sensitive period 1950 to 2014, and discuss the aerosol trend and its
relation to the increase in the radiative imbalance as seen from space.

## 2 Method

In this work ERF and climate sensitivity are estimated using a Bayesian framework with prior estimates of historical
anthropogenic and natural ERF and historical observations of surface temperature and ocean heat content (OHC).

### 2.1 Bayesian estimation model

The Bayesian estimation model was first documented in Aldrin et al. (2012) and further developed in Skeie et al. (2014,
2018). The full model consists of a dynamic process model with an idealized representation of the Earth's energy balance - a
simple climate model (hereafter referred to as the SCM), a data model that describes how various observations are related to
the process states, and a parameter model that expresses our prior knowledge of the parameters. The model is described in
detail in Appendix A. The method simultaneously estimates posterior estimates (including uncertainties) of the ERFs, the
$ECS_{inf,}$ and other model parameters.



## 2.2 Effective Radiative Forcing

The prior time series for the ERF used in this study is from the IPCC AR6 report (Smith et al., 2021a;Forster et al., 2021) and the extension of these forcing time series to 2022 (Forster et al., 2023) hereafter named AR6 extended.

The IPCC AR6 provides the best estimate and the 5th and the 95th percentile of the forcing time series. The aerosol ERF is separated into ERF due to aerosol-radiation interactions (ERFari) and aerosol-cloud interactions (ERFaci). Table A1 lists the forcing components included in our estimation. The implementation of the ERF uncertainties from AR6 and AR6 extended is described in Appendix A and the ERF priors are shown in Fig. S1 and Fig. S2.

## 2.3 Observational data

The observational-based data series used in the Bayesian estimation model are annual hemispheric means of surface temperature (blended sea surface temperature over ocean and 2-metre temperature over land), global annual OHC 0-700 meter and 700-2000 meter as well as an ENSO-index. Three different data series for surface temperature and seven different data series for OHC are used. As an ENSO-index the monthly Southern Oscillation Index (SOI) is used. The observation-based data series used are listed in Table A2 and shown in Fig. S3-5.

In the Bayesian estimation we use the temporal evolution of the reported errors for the observation-based series and estimate their magnitudes within the model, taking into account the possibilities that the reported standard errors may under- or overestimate the true uncertainty. For OHC, we choose to use the temporal evolution of the uncertainty from one dataset, as the reported uncertainties (Fig. S6) may not include the full uncertainties in OHC (see Appendix A).

As the representation of deep-water formation in the SCM is simplified, putting heat at the bottom layer of the model, we compare the OHC for 700 to 2000 meter in observations to OHC below 700 meter in the model. We do not include observations of OHC data below 2000 meters due to limited observations time series and assume these within the uncertainties in the observed OHC.

## 2.4 Estimations

As a starting point, we use the baseline estimation from Skeie et al. (2018), where the IPCC AR5 time series (Myhre et al., 2013a) were used as the ERF priors and observations up to 2014 were included ("*Skeie18*"). We replace the AR5 prior with the AR6 prior and further with the AR6 extended prior. In Fig. S1 the AR5, AR6 and AR6 extended ERF priors are compared. IPCC AR5 did not provide estimates of how the uncertainties evolved in time and in Skeie et al. (2018) we assumed that the uncertainties in ERF scaled with time and not magnitude of the forcing as in AR6. Updating the ERF prior, extending up to 2019 ("*Base*") and 2022 ("*Base extended*") respectively, more recent observations could also be included. When forcing priors are replaced and additional years with observations included (Table 1), this is done stepwise as described in Appendix A and summarized in Table A3.



For each ensemble member in the Bayesian estimation, the sampled uncertainty scale factor for each ERF component is applied to the whole time series. There are hence no uncertainties in the time evolution of the ERF, identified as a limitation of the method (Skeie et al 2018). Therefore, a range of sensitivity tests are done for the setup in "Base", where aerosol ERFs from AR6 are replaced with idealized alternative pathways. Four sets of sensitivity tests are performed where the priors for ERFari and ERFaci are adjusted (Fig. S7 and Fig. S8). First, we perform a sensitivity test where the aerosol ERFs are

smoothed. Thereafter we perform a second group of sensitivity tests where aerosol ERFs in different time periods are adjusted. In the third group of sensitivity tests we change the year of the strongest aerosol ERF by replacing the AR6 aerosol ERF from 1950 to a chosen year by a linear ERF. The fourth and final set of sensitivity tests is similar to the third group of sensitivity tests, but the aerosol ERFs are kept constant for the years following the end of the linear change. The entire aerosol ERF time series is scaled, so the 2019 aerosol ERF prior distribution is equal to the AR6 distribution. The baseline

for the two latter groups of sensitivity tests are the smoothed aerosol ERFs. We compare the different sensitivity tests and the baseline estimation to see the effect of different aerosol ERF pathways on the posterior estimates of $ECS_{inf}$ and ERF.

Finally, based on results from the sensitivity tests of adjusting the aerosol ERF in different time periods, we do a test where ERFaci in 1950 and 2014 are independent of each other. We draw from the distribution of ERFaci in 1950 and use this scaling factor prior to 1950. Similarly, we draw from the distribution in 2014 and use that scaling factor thereafter. In the

period between we linearly interpolate these two scaling factors. The rate of change in ERFaci prior has therefore much larger variability than in the baseline setup.

The setup and stepwise update of the baseline estimations as well as all the sensitivity tests are summarized in Table A3.

**Table 1: List of estimations performed with a description of the setup, ERF prior used and end year. The description of the**
**stepwise change in data and priors used for estimations and all the sensitivity tests are given in Table A3.**

| Simulation | Description | ERF prior | End year |
|---|---|---|---|
| *"Skeie18"* | The main analysis in Skeie et al. (2018) | AR5 | 2014 |
| *"Base"* | Base simulation with AR6 prior | AR6 | 2019 |
| "*Base extended*" | Base simulation with AR6 extended | AR6 extended | 2022 |
| Sensitivity tests | ERFari and ERFaci from AR6 are modified in specific time periods (see description of each test in Table A3). | AR6 | 2019 |
| Sensitivity test: ERFaci trend | ERFaci uncertainties in 1950 (and before) and 2014 (and after) independent of each other. | AR6 extended | 2022 |



## 3 Results

In this section we present constrained estimates of $ECS_{inf}$, anthropogenic ERF and aerosol ERF using observations of OHC and surface temperature and prior forcing time series. First, we use observations and ERF up to 2019 and thereafter update

our analysis based on data including the year 2022. We investigate the sensitivity of the posterior estimates to the prior aerosol ERF temporal evolution, and finally let the ERFaci more freely evolve within the uncertainty range.

### 3.1 Estimations using AR6 and AR6 extended ERF time series

The starting point for the analysis is the results from Skeie et al. (2018). For "*Base*", when the AR5 forcing prior is replaced by the AR6 prior and estimation includes data up to 2019 (Table 1), the $ECS_{inf}$ posterior mean is 2.1 K and the 90%

confidence interval (C.I.) is 1.5 to 2.9 K. The posterior mean $ECS_{inf}$ is 0.2 K higher and the 90% C.I. is narrower compared to "*Skeie18*" that only included data up to 2014.

We further extend the analysis up to 2022 using the extended AR6 ERF time series from Forster et al. (2023) (Fig. S1 and Fig. S2) and updated and extended observational time series (Table A2, Fig. S3 and Fig. S5). For this set up ("*Base extended*") the $ECS_{inf}$ estimate shift to slightly larger values compared to "*Base*" with a mean estimate of 2.2 K and the 90%

C.I. ranging from 1.5 to 3.0 K (Fig. 1). Each step for updating and extending the data are described Table A3, and the resulting posteriori $ECS_{inf}$ presented in Table S1.



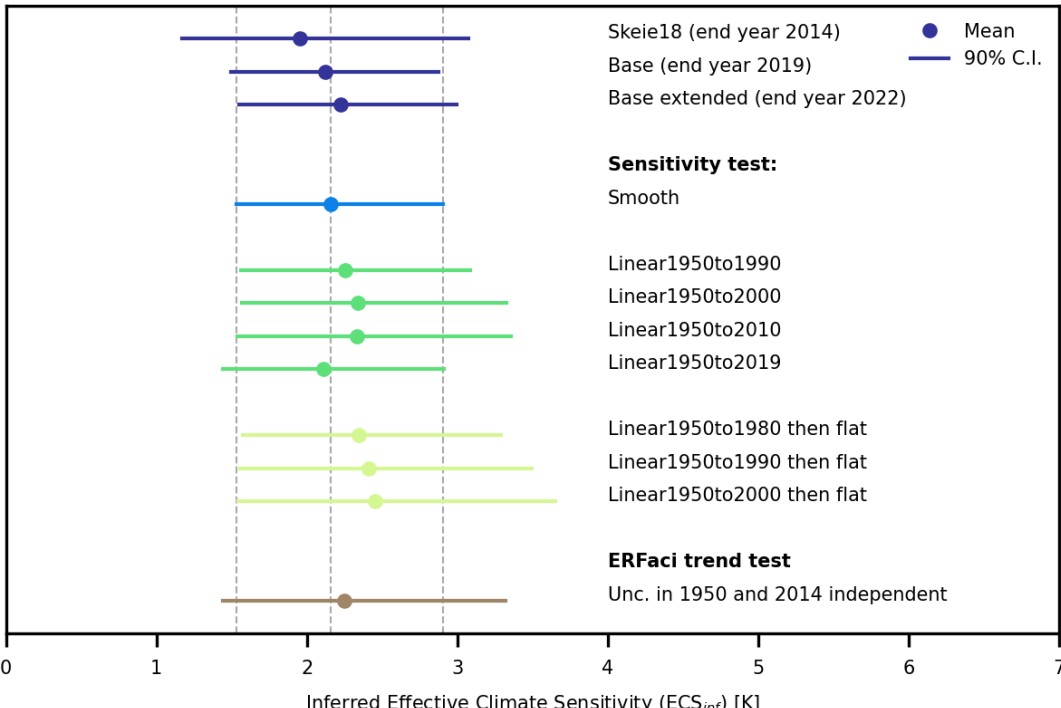

Inferred Effective Climate Sensitivity ($ECS_{inf}$) [K]

**Figure 1: Posterior inferred effective climate sensitivity ($ECS_{inf}$) for the different analyses and sensitivity tests. The 5th to 95th percentiles ranges are indicated by a solid line and the mean values as a dot. The vertical lines indicate the posterior mean value and the 90% C.I. for "*Smooth*", as it is used as a starting point for the "Linear" sensitivity tests shown. The posterior mean, median and 5th and 95th percentiles are presented in Table S1 and description of all estimations in Table A3. For AR5 we use the best estimate of 2xCO2 radiative forcing from AR5, while for AR6 we use the 2xCO2 ERF corresponding to the posteriori estimate of the historical $CO_2$ ERF for the conversion from climate sensitivity parameter to $ECS_{inf}$ (see Appendix A).**

For the transient climate response (TCR), the posterior mean is 0.1 K higher using AR6 forcing as a prior and observations up to 2019 compared to "*Skeie18*" with data up to 2014 (Fig. S9). In "*Base*", the mean value is 1.5 K, and the 90% C.I. is 1.1 to 2.0 K. Extending the analysis up to 2022 resulted in a slight increase in the TCR with mean value of 1.6 K and the 90% C.I of 1.1 to 2.1 K (Fig. S9, Table S2).

The prior and posterior distribution of the anthropogenic ERF are shown in Fig. 2. For "*Skeie18*" with AR5 forcing prior, the posterior and prior distributions were similar in 2014, while in "*Base*" the prior and posterior distributions are quite different (Fig. 2a). The prior ERF distribution in 2014 is similar for AR5 and AR6 (Fig. 2a), while the time evolution of the prior is quite different (Fig. 2c). From the stepwise update and extension of the data used in the estimation (Table S3), the temporal evolution of the forcing pathway seems to play a large role in explaining why the prior and posterior distribution of the anthropogenic ERF for the end year are so different using AR6 forcing prior and similar using AR5 forcing prior (Fig. 2b).





For "*Base*", the posterior mean is 3.1 W m$^{-2}$ with a 90% C.I. from 2.6 to 3.7 W m$^{-2}$, where the lower limit is close to the prior mean of 2.7 W m$^{-2}$ from AR6 (Table S3).

Extending the analysis further to 2022, using AR6 extended forcing prior, resulted in a similar posterior mean for anthropogenic ERF of 3.2 W m$^{-2}$ and 90% C.I. from 2.7 to 3.7 W m$^{-2}$ as for 2019 using the AR6 prior ("*Base*"). The
posterior distribution of anthropogenic ERF in "*Base*" (for 2019) and "*Base extended*" (for 2022) were similar despite the prior distribution being shifted to larger values in the latter (Fig. 2b).

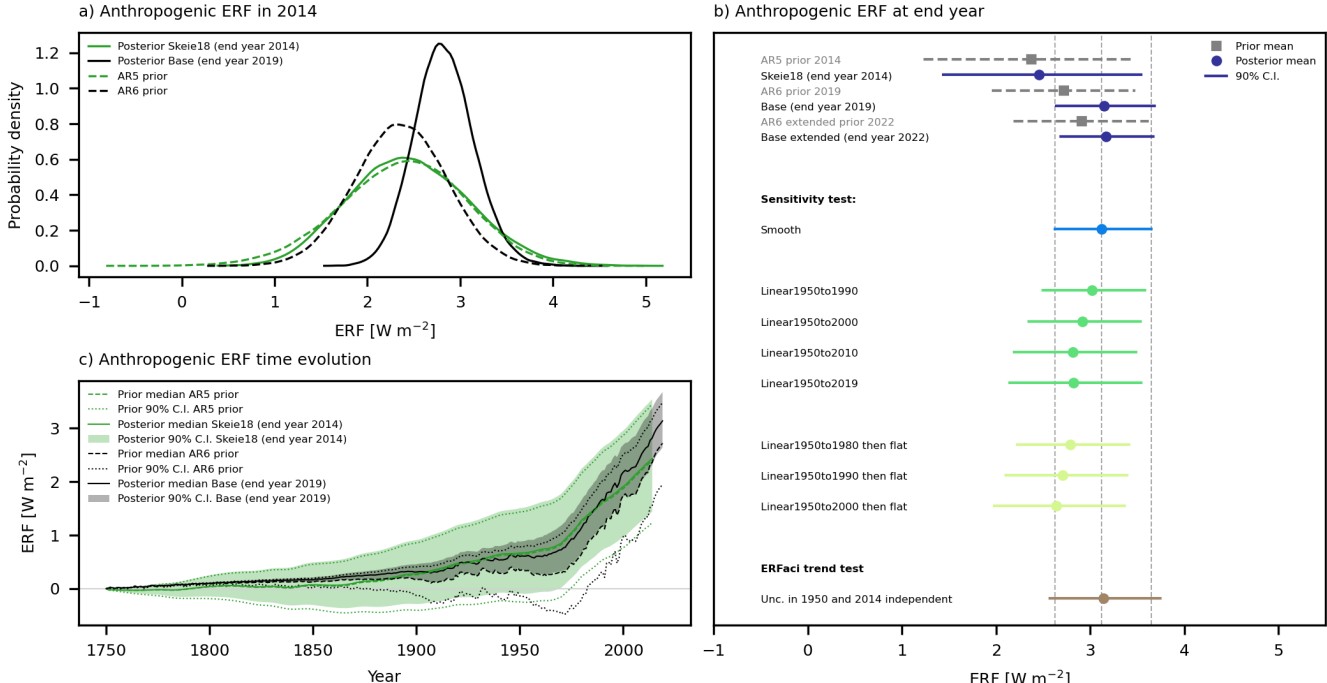

**Figure 2: Prior and posterior distributions of anthropogenic ERF. In a) the probability density function of the prior (dashed line) and posterior (solid line) ERF in 2014 for "*Base*" using AR6 prior (black) and "*Skeie18*" using AR5 prior (green). In b) the posterior 90% C.I. is indicated by a solid line and the posterior mean as a dot and the prior 90% C.I. is indicated by a dashed line and the prior mean as a square. The results are shown for the end year in the analysis. In c) the prior and posterior time evolutions of the anthropogenic ERF are shown for "*Skeie18*" and "*Base*". The underlying numbers for b) are presented in Table S3.**

Looking at all the individual forcing components, the "*Base*" prior and posterior distributions of the ERF in 2019 are similar for all components except for ERFaci (Fig. S10). This points to aerosols for the difference in the prior and posterior anthropogenic ERF (Fig. 2). Figure 3 shows the prior and posterior distribution of the total aerosol ERF, the sum of ERFaci and ERFari. In "*Base*", the posterior values for 2019 are shifted to weaker values and the distributions are narrower compared to the AR6 prior (Fig. 3b) with a posterior mean of -0.68 W m$^{-2}$ and 90% C.I. ranging from -1.1 to -0.28 W m$^{-2}$



(Table S4). Stronger aerosol ERF than the prior mean of -1.1 W m⁻² in 2019 is not supported by this analysis. Also, for "*Base extended*", the posterior distribution for aerosol ERF in 2022 is mostly in the weaker half of the prior distribution (Fig. 3b).

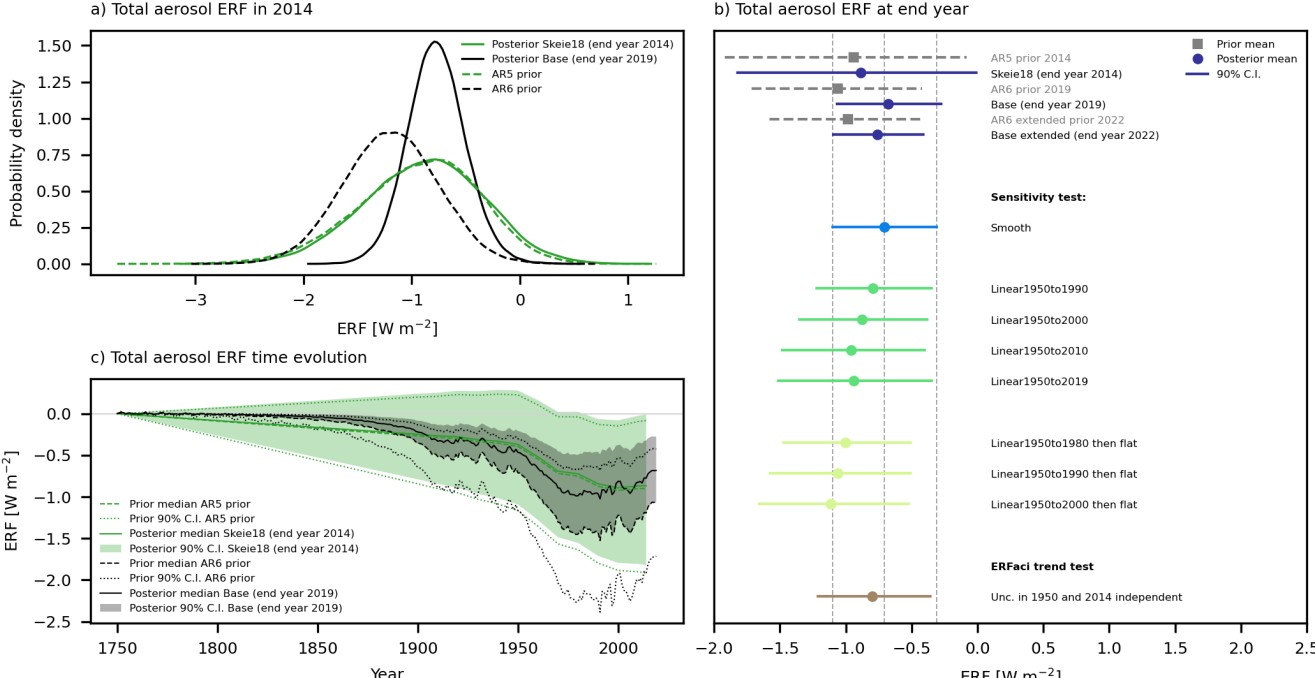

**Figure 3. Same as Fig. 2, but for aerosol ERF. The underlying numbers for b) is presented in Table S4.**

However, the prior time evolution in AR5 and AR6 are quite different for aerosol ERF (Fig. 3c). In AR5 the aerosol ERF strengthened gradually from 1950 to around 2000 and was quite constant thereafter. In AR6 the aerosol ERF shows a steep strengthening from 1950s to 1970s, then the aerosol ERF is quite stable for some decades with some interannual variability, and from around 2005 the aerosol ERF is weakening in magnitude. Looking at the time evolution of the 5th percentile of

total anthropogenic ERF (Fig. 2c) for "*Base*", the prior shows a reduction in anthropogenic ERF from the 1950s until the 1970s, while in the posterior the ERF is flat over this period. The observed temperature and OHC used in the estimation do not allow for a weakening in the anthropogenic ERF in the second half of the century. A possible explanation of the weak aerosol ERF for the end year in "*Base*" and "*Base extended*" compared to the prior is that the observations do not allow for aerosol ERF in the stronger range earlier in the period (1970-1990s) and hence exclude ERF in the stronger range later in the

period as well, as the uncertainty is represented as a fixed factor for the entire historical period.



## 3.2 Sensitivity test for aerosol ERF evolution

In the setup for "*Base*" and "*Base extended*", no uncertainties in the time evolution of the ERF time series are included. For each ensemble member, a fixed scaling factor is used for the entire period. For both "*Base*" and "*Base extended*", the posterior aerosol ERF ended up in the weaker part of the prior distribution for 2019 and 2022 respectively (Fig. 3b). The possible explanation outlined above is that the observations do not allow for a strong aerosol ERF earlier in the period, and therefore not in the later part of the period either. To investigate this potential explanation further we perform four sets of sensitivity tests where the AR6 aerosol ERF prior is replaced by idealized ERF priors (Table A3).

The prior for aerosol ERF shows some year-to-year variability (Fig. 3c). Therefore, the first test is to replace the aerosol ERF from AR6 with a smoothed forcing time series (Fig. S7a) to see the effect of the year-to-year variability in the forcing prior on the posterior estimates. With a smoothed aerosol ERF prior, the posteriori distributions of $ECS_{inf}$, TCR, aerosol ERF and anthropogenic ERF are similar to "*Base*" (Fig. 1, Fig. S9, Fig. 3b, Fig. S11a, Fig. 2b, Fig. S12a). This indicates that the posterior estimates are not sensitive to interannual variability in the forcing prior.

The second group of sensitivity tests adjusts the aerosol ERF in different time periods to identify periods where the posteriori estimates are sensitive to the aerosol forcing time evolution (Fig. S7 b). Adjusting the time series prior to 1950 had minor effect on the posteriori estimates of $ECS_{inf}$ (Table S1). Also adjusting the aerosol ERF from 2014 led to only minor changes in the posterior $ECS_{inf}$ compared to "*Base*". However, adjusting the aerosol forcing pathway from 1950 and onwards had a greater impact on the posteriori estimates. For the two periods 1950 to 1980 and 1980 to 2019, the difference for the 95$^{th}$ percentile of $ECS_{inf}$ between the two tests performed were approximately 0.5 K for both periods (Table S1).

The third group of sensitivity tests investigate the sensitivity of shifting the year of the strongest aerosol ERF to later years. The aerosol ERF evolution from 1950 to a given year was replaced by a linear strengthening of the aerosol ERF (Fig. S7c). The smoothed AR6 aerosol ERF prior ("*Smooth*") was used as the starting point here, as the aerosol ERF in AR6 shows large interannual variability in the early 21$^{st}$ century. The posteriori present-day aerosol ERF strengthened as the aerosol ERF maximum was shifted from around 1980 as in "*Smooth*" to 1990, 2000 and 2010 (Fig. 3b) and accordingly the present-day anthropogenic ERF weakened (Fig. 2b). The posteriori $ECS_{inf}$ increased from 2.1 K in "*Smooth*" to 2.3 K in these sensitivity tests (Fig. 1). With a linear reduction of aerosol ERF from 1950 and all the way to 2019, the posteriori distribution of aerosol ERF in 2019 is similar to "*Linear1950to2010*" (Fig. 3b), but the estimated $ECS_{inf}$ is quite different where the posterior mean is reduced by 0.2 K (Fig. 1). Due to the shape of the aerosol ERF history, the integrated posterior aerosol ERF between 1950 and 2019 is weaker in "*Linear1950to2019*" compared to "*Linear1950to2010*" (Fig. S11c) and correspondingly the integrated total anthropogenic ERF is stronger in "*Linear1950to2019*" compared to "*Linear1950to2010*" (Fig. S12c). This further highlights the importance of the time evolution of the prior ERF and not only the present-day ERF value for constraining $ECS_{inf}$.

The final set of idealized sensitivity tests is similar to the linear sensitivity test, but here the aerosol ERF is kept constant for the period following the linear strengthening of the forcing. The full aerosol ERF time series is then scaled so the aerosol



ERF distribution in 2019 is similar as in AR6 (Fig. S7d). These aerosol ERF time series represent a saturation of the aerosol ERF after the strongest ERF is reached. In these sensitivity tests the aerosol ERF stabilizes over the recent decades, like the AR5 prior where the aerosol forcing stabilized after the 1990s (Fig. 3c). The later the stabilization period starts, the stronger the posteriori aerosol ERF in 2019 (Fig. 3b) and larger $ECS_{inf}$ and TCR (Fig. 1, Fig. S1). For the test with a linear reduction of aerosol ERF from 1950 to 2000 and a constant value thereafter, the posteriori distribution of aerosol ERF for 2019 is

similar to the AR6 prior (Fig. 3b) and the posteriori mean $ECS_{inf}$ is 2.5 K and 95th percentile 3.6 K.

**3.3 Sensitivity test adjusting ERFaci trend**

Based on the results from the sensitivity tests replacing the AR6 ERF time series with idealized aerosol ERF pathways in specific time periods, we do a test where the aerosol ERF trend is allowed to vary more than in the baseline setup. We build this test on "*Base extended*" using data up to and including 2022. We draw from the uncertainty distribution of ERFaci in

2014 and 1950 independently. Prior to 1950 and after 2014 we use the respective scaling factors, while for the period between 1950 and 2014 we linearly interpolate these two factors. An ensemble with a weak ERFaci in 1950 and a strong ERFaci in 2014 will have a different temporal pathway than a strong ERFaci in 1950 and a weak ERFaci in 2014.

The $ECS_{inf}$ posteriori mean was similar but the 90% C.I. widened in this test compared to "*Base extended*", with an upper value of 3.3 K compared to 3.0 in "*Base extended*" (Fig. 1). Also, the posteriori estimates of TCR with a 90% C.I. of 1.1 to

2.2 K were wider compared to "Base extended" with 0.1 K higher upper limit of the 90% C.I. (Fig. S1).

The posterior mean aerosol ERF in 2022 was -0.80 W m$^{-2}$ with a 90% C.I. of -1.2 to -0.36 W m$^{-2}$ compared to the prior mean of -0.98 W m$^{-2}$ and 90% C.I. of -1.6 to -0.41 W m$^{-2}$ (Fig. 3b). Still, the strongest aerosol ERF in AR6 extended for 2022 is not supported by this analysis even though the aerosol ERF in the latter half of the 20th century is not directly tied to the ERF in 2022.

**4 Discussion**

Although we test our estimation method with a wide range of highly idealized aerosol ERF pathways, our posteriori estimates of the climate sensitivities (Fig. 1) are weaker than community estimates of climate sensitivity (Forster et al., 2021;Sherwood et al., 2020) and estimates from most climate models. 21 out of 42 CMIP6 models have climate sensitivities larger than the maximum 95th percentile of 3.6 K (Fig. 1) (Smith et al., 2021a). In the following discussion, we will first

relate our estimates of $ECS_{inf}$ to the climate models' climate sensitivities. Then we will discuss the influence of the aerosol ERF pathway on our estimates and at the end how the results here relate to Earth's Energy Imbalance (EEI) as observed from space.





## 4.1 ECS$_{inf}$ versus ECS

In this work we estimate the inferred effective climate sensitivity based on historical observations of temperature and OHC.

For calculations of climate sensitivity in climate models, the models are run with an abrupt quadrupling of the $CO_2$ concentrations from pre-industrial levels. To equilibrate the models requires to run the model for thousands of years (Rugenstein et al., 2020). The climate models are instead run for a shorter period, often 150 years, and surface temperature anomalies and top of atmosphere (TOA) net downwelling radiative flux anomalies regressed to project the climate sensitivity, a method developed by Gregory et al. (2004) and used for CMIP6 models in Zelinka et al. (2020). This climate

sensitivity is termed effective climate sensitivity (ECS) and does not include feedbacks occurring over longer timescales than 150 years. The ECS$_{inf}$ calculated here will differ from the climate models' ECS for several reasons which we will go through below.



**Figure 4: Illustrations of reasons for differences between observational estimates of ECS$_{inf}$ and Effective Climate Sensitivity (ECS) as diagnosed from climate models. In a) the influence on ECS by different definitions of temperature is illustrated. The ECS$_{inf}$ estimated in "*Base*", based on observations of GMST, are enhanced by 4% and 10% as changes in GMST are larger than GSAT in climate models. In b) different assumptions of the "Pattern Effect" are added to the ECS$_{inf}$ estimate from "*Base*". The different values for $\alpha'$ is given in the legend. For "*Base*" $\alpha'$=0. In c) the results are summarized by the mean value (filled circle), median value (x) and 5$^{th}$ to 95$^{th}$ percentile range as solid line. Also indicated is the assessed best estimate of Equilibrium Climate Sensitivity from IPCC AR6 (filled circle) and the very likely (solid line) and likely (dashed line) uncertainty range.**

First, the observed time series for global mean temperatures are blended products of measured air temperature (2-metre above surface) over land and sea surface temperature over ocean, termed global mean surface temperature (GMST). In the SCM this definition of temperature change is implemented, while in climate modelling, the global mean surface air temperature (GSAT) is used for calculation of ECS. In IPCC AR6 the difference between the long-term change in GSAT and GMST was assessed to be less than 10%, but with low confidence in the sign of the difference (Gulev et al., 2021a) as GSAT increases faster than GMST in climate models while a limited numbers of observational studies show the opposite. To illustrate the effect of the definition of temperature on the ECS$_{inf}$, the probability density function of ECS$_{inf}$ in "Base" is enhanced by 4 and 10% (Fig. 4a). We only illustrate an increase here as changes in GSAT are larger than GMST in climate models. For a 10% increase in the temperature response, the ECS$_{inf}$ median value increased from 2.1 to 2.3 K (Fig. 4c).

Both the historical $CO_2$ ERF and the ERF for a doubling of $CO_2$ concentration (2xCO2 ERF) are assessed with uncertainties (Forster et al., 2021), and there is considerable spread in $CO_2$ ERF diagnosed in climate models (Smith et al., 2020). The difference in $CO_2$ forcing in the climate models has implications for ECS diagnosed from the model simulations (Cess et al., 1993;Soden et al., 2018), as a larger forcing results in larger temperature response. For illustration, for the 90% uncertainty range of 2xCO2 ERF from IPCC AR6 (3.46-4.40 Wm$^{-2}$) and a best estimate of the climate sensitivity parameter of 0.76 K (W m$^{-2}$)$^{-1}$ (3K/3.93 W m$^{-2}$) (Forster et al., 2021) the ECS will increase or decrease by 0.36 K (36%) relative to the best estimate of 3 K. In addition, the forcing strength can also change as the climate changes. The instantaneous radiative forcing of $CO_2$ is found to be dependent on the climatic base state and increases by 25% for every doubling of $CO_2$ (He et al., 2023). This contributes to a ~15 to 20% increase in climate sensitivity for every doubling of $CO_2$. ECS diagnosed in climate models from 4 times $CO_2$ concentration will hence be larger than ECS$_{inf}$ derived from observations over the historical period where $CO_2$ concentration only has increased by ~50%.

The strength of the climate feedbacks can also change over time (Armour, 2017;Senior and Mitchell, 2000), and one reason for this is the so called "Pattern Effect" (Stevens et al., 2016). The radiative feedbacks depend on the spatial pattern of the warming (e.g. Andrews et al., 2015), and as the spatial pattern of surface temperature evolves over time, the climate feedbacks change.

To take this time dependency of the feedbacks into account, the equilibrium climate sensitivity can be written as $ECS = -\Delta F_{2xco2}/(\alpha + \alpha')$ where $\alpha$ is the effective feedback parameter estimated over the historical period and $\alpha'$ represents the change in the feedback parameter between the historical period and the time of equilibrium for a 2xCO2 forcing ($\Delta F_{2xco2}$). The $\alpha'$ factor can be calculated from Earth System Models and IPCC AR6 assessed $\alpha'$ to be in the range of 0.0 to 1.0 W m$^{-}$





$^2$ K$^{-1}$ (Forster et al., 2021). The feedback parameter $\alpha$ [W m$^{-2}$ K$^{-1}$] quantifies the change in net energy flux at the TOA for a given change in global temperature and represent the Planck response and all other feedbacks. In the SCM, the climate sensitivity parameter $\lambda$ [K [W m$^{-2}$]$^{-1}$] represents these feedbacks. The relationship between the effective $\lambda$ and $\alpha$ (as estimated based on observations over the historical period) is $\lambda = -\frac{1}{\alpha}$. To test the effect of changes in the climate feedback

over time we estimate ECS from an adjusted climate sensitivity parameter $\lambda + \lambda' = -\frac{1}{\alpha + \alpha'} = \frac{\lambda}{(1 - \lambda \alpha')}$. For the posteriori estimates of $\lambda$ we convert to $\lambda + \lambda'$ for four different values of $\alpha'$ (0.1, 0.3, 0.5, 1.0) in addition to $\alpha'$ equal to zero as in "*Base*", spanning the range of 0.0 to 1.0 W m$^{-2}$ K$^{-1}$ from IPCC AR6. The probability density functions are then shifted to larger values, stretching the tail towards higher values for the climate sensitivity (Fig. 4b). A pattern effect of 0.5 W m$^{-2}$ K$^{-1}$ shifts the ECS$_{inf}$ mean value in "*Base*" to 3 K, which is the best estimate of the ECS assessed by IPCC AR6. A stronger

pattern effect of 1 W m$^{-2}$ K$^{-1}$ gives much larger climate sensitivity estimates with a mean value of 6.4 K (Fig. 4c) but note that the values of $\lambda$ now extends our prior range.

The observed sea surface temperature pattern, with a stronger warming in the western Pacific and a cooling in the eastern Pacific, is not simulated within coupled atmosphere-ocean climate models (Fueglistaler and Silvers, 2021;Wills et al., 2022). Internal natural variability, as represented in the model, are unlikely to be the reason for the observed temperature in the

Pacific, and Wills et al. (2022) points at model biases in the response to historical forcing as part of the discrepancy. Recently, using an ensemble from a single climate model with idealized step changes in aerosol emissions, Hwang et al. (2024) found that the equatorial Pacific cooled following an increase in aerosols emissions and that the cooling persist for several decades after the aerosols were removed. Limitations in our understanding of the drivers of the observed SST patterns and uncertainty in the strength of the "real world" pattern effect and how it will evolve in the near future,

substantially increases the uncertainty about future warming (Zhou et al., 2021).

Although the temperature definition and CO$_2$ ERF can explain some of the differences between climate sensitivity calculated from climate models and inferred from observations, the assumptions on the Pattern Effect are crucial. Pattern Effects within the IPCC AR6 assessed range of 0 to 1 W m$^{-2}$ K$^{-1}$ can shift the mean value of ECS$_{inf}$ to values larger than the upper very likely limit of 5 K from IPCC AR6. The upper limit of climate sensitivity cannot be constrained by historical observations

due to the Pattern Effect by definition. The recognition of the pattern effect reconciled the previous discrepancies of historical constrained climate sensitivity estimates and ECS in climate models (Forster et al., 2021), and the pattern effect is a limiting factor in the quest for constraining the "true" climate sensitivity.

The TCR should be less influenced by the pattern effect. In "*Base extended*" the posteriori mean is 1.6 K, with 90% C.I. of 1.1 to 2.1 K. The posteriori distribution of TCR are weaker compared to the IPCC AR6 assessment (Forster et al., 2021)

where the 90% C.I. includes values lower than the very likely lower bound of 1.2 K from IPCC AR6, but not values greater than the likely upper bound of 2.2 K from IPCC AR6 (Forster et al., 2021) (Fig. S9).



## 4.2 Aerosol Forcing trend

In "*Base*", the posteriori distribution of the present-day (2019 relative to 1750) aerosol ERF was shifted to weaker values compared to the prior based on IPCC AR6 with a very likely range of –1.7 to –0.4 W m$^{-2}$. The uncertainty in aerosol ERF in
"*Base*" was not allowed to change over time. In the estimation, updating the aerosol forcing values based on observed temperature change and OHC, a strong aerosol forcing from the mid-20$^{th}$ century was prohibited (Fig. 3c) and hence a strong present-day aerosol forcing (Fig. 3b) and a weak present-day anthropogenic ERF (Fig. 2b) were excluded.

From the sensitivity test on historical aerosol time evolution, the posteriori distribution of the present-day aerosol ERF can be very different for different assumptions on the aerosol ERF time evolution (Fig. 3b). This underlines the importance of
the pathways of the aerosol ERF, not only for this approach of estimating ECS$_{inf}$ and ERFs based on historical observations, but also for climate models evaluated on historical changes and used for projections of future climate evolution. As also highlighted in Smith and Forster (2021), if the forcing is not correct, the temperature in the past and projections for the future will be biased. The climate models contributing to RFMIP and AerChemMIP showed quite variable time evolution of the diagnosed aerosol ERF (Smith et al., 2021b). Using an emission to forcing relationship based on the diagnosed aerosol ERF
in these CMIP6 models and a climate emulator trained on CMIP6 models, Smith et al. (2021b) constrained the aerosol ERF using observed near-surface warming and the 1971 to 2018 Earth energy uptake. They estimated aerosol ERF in 2019 with a slightly narrower uncertainty range than IPCC AR6 of −1.5 to −0.4 W m$^{-2}$, and a modest recovery in aerosol forcing (+0.025 W m$^{-2}$ decade$^{-1}$) between 1980 and 2014. We find a slightly larger change in aerosol forcing of +0.031 W m$^{-2}$ decade$^{-1}$ over the same period in "*Base extended*", slightly weaker than +0.035 W m$^{-2}$ decade$^{-1}$ in the prior. If we then
let the ERFaci more freely evolve within the AR6 uncertainty range between 1950 and 2014, the trend in aerosol ERF is similar to Smith et al. (2021b) with +0.024 W m$^{-2}$ decade$^{-1}$ between 1980 and 2014.

In Fig. 5, the 20-year linear trend for six different periods after 1950 are shown for "*Base extended*" and the test where ERFaci more freely evolves within the AR6 uncertainty range. As the aerosol ERF prior is quite stable from the 1970s to early 2000s (Fig. S1) and the uncertainties are tied to the uncertainty in the end year, the prior as well as the posterior for the
aerosol ERF trend is narrow and close to zero in this period. When the ERFaci is allowed to evolve more freely within the IPCC AR6 range, both the prior as well as the posterior 90% C.I. for the aerosol ERF trend are broadened (Fig. 5). For the earliest two periods, 1950-1969 and 1960-1979, and the latest period, 2000-2019, the posterior trends are weaker than in the prior. The posterior 2000-2019 aerosol ERF trend is in the weaker range of Smith et al. (2021b) and similar to Albright et al. (2021) with increased variance, both presented in Quaas et al. (2022). Keep in mind that in this test, the ERFari is still tied to
the AR6 ERFari pathway, as is ERFaci post 2014. For comparison, the median aerosol ERF trend of 0.1 W m$^{-2}$ decade$^{-1}$ for the 2000 to 2019 period is a third of the increase in CO$_2$ ERF over the same period.

Also shown in Fig. 5 are trends calculated from aerosol ERF diagnosed within CMIP6 models (Smith et al., 2021b). The spread in trend is large in all periods, and for the 1960 to 1979 period the modeled ERF trends are in the weaker range of the



priors and more consistent with our posterior distributions. In the 1980 to 1999 period most of the CMIP6 models have a

negative aerosol ERF trend, while the posteriori mean is close to zero, but the 90% C.I, is wide.

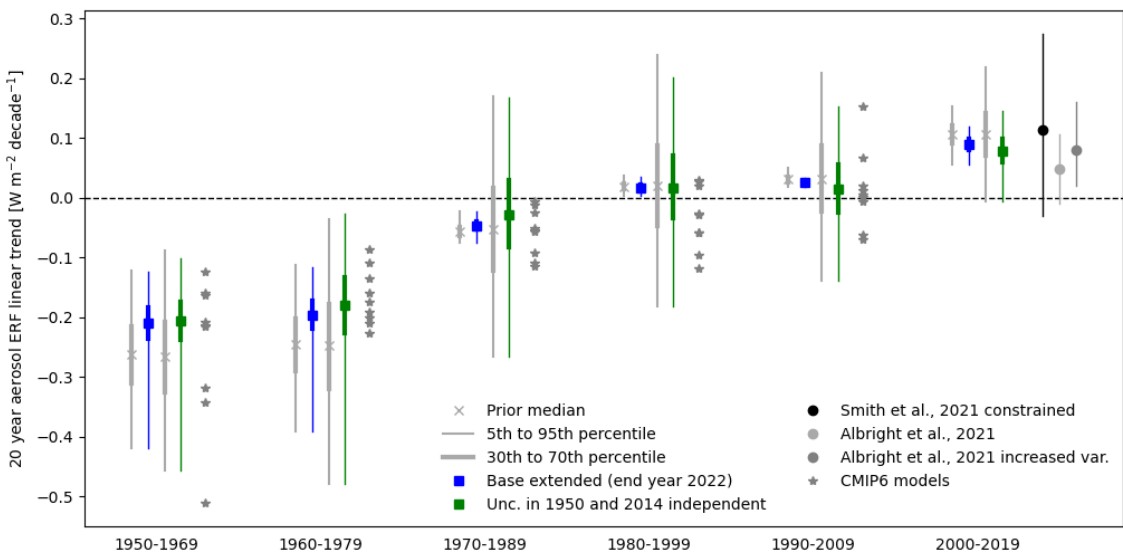

**Figure 5: The 20-year aerosol ERF linear trend for six different periods. The posterior aerosol ERF trends are shown for "*Base extended*" (blue) and the test with independent uncertainties in 1950 and 2014 (green) with corresponding prior distribution plotted in gray to the left of the posterior. The median (crosses for prior and squares for posterior), 30th to 70th percentile and 5th**

**to 95th percentiles are shown. For the five first time periods, the linear trend in the ERF diagnosed from 9 CMIP6 models in Smith et al. (2021b) are shown as stars and the linear trends are calculated on smoothed time series of ERF. For the last period (2000 to 2019) the mean, 5th and 95th percentiles presented in Quaas et al. (2022) from Albright et al. (2021) and Smith et al. (2021b) are shown.**

### 4.3 Energy Imbalance

The trend in the Earth's Energy Imbalance (EEI) can be estimated based on satellite retrievals (Loeb et al., 2018a) and can in

principle give additional information of the aerosol ERF pathway. The EEI is the net radiative flux at the top of atmosphere

(TOA), and it determines the evolution of global temperature change. For a positive imbalance at the TOA, less energy is

leaving than entering the system, heat is stored in the system and surface temperature will increase to restore energy balance

at TOA. The EEI is the portion of the radiative forcing that has not yet been responded to (Hansen et al., 2005). In a linear

framework this can be written as: $EEI = ERF + \alpha\Delta T$, where $\alpha$ is the net total feedback parameter [W m$^{-2}$ K$^{-1}$], which

represents the combined effect of the various climate feedbacks. In this study the climate sensitivity parameter $\lambda = -1/\alpha$

represents these feedbacks.

From the TOA Earth radiation budget data from the Clouds and the Earth's Energy System (CERES) data, the trend in EEI

can be analyzed (Loeb et al., 2018a). As the absolute values of the radiation fluxes are too uncertain, these are anchored to

the mean of observed rate of heat gain, mainly storage of heat in the ocean over a reference period (2005 to 2015) (Loeb et



al., 2018b). The trend in CERES data can be used as additional information in our Bayesian set up, as the trend in EEI is independent of the OHC data that are already included in our estimation. Here, however, we only compare the posterior estimates of EEI with the CERES EEI trend.

Fig 6a shows the EEI from "*Base extended*" from the forced temperature response as well as the temperature response
including ENSO variability. Including ENSO, the median value averaged over the period 2006 to 2020 of 0.74 W m$^{-2}$ is in good agreement with the 0.76±0.2 W m$^{-2}$ from von Schuckmann et al. (2023), which also holds for the longer period 1971 to 2020 with median value of 0.49 W m$^{-2}$ in this study and 0.48 W m$^{-2}$ in von Schuckmann et al. (2023). This is as expected as observational based time series of OHC used in this study are included in the assessment of von Schuckmann et al. (2023) (see also Fig. S4). For the trend in EEI, the forced EEI with and without the ENSO response, are 0.2 W m$^{-2}$ decade$^{-1}$
from 2005 to 2022. This is clearly weaker than the linear trend for the 12-month running mean CERES EEI of 0.44 W m$^{-2}$ decade$^{-1}$ (Fig. 6a). Adding the ENSO temperature response enhanced the year-to-year variability in EEI, but the variability is weaker than in CERES (Fig. 6a) that show pronounced interannual variability driven primarily by clouds (Loeb et al., 2018a;Loeb et al., 2021), a variability a SCM would not capture.

Although the posteriori results of EEI averaged over the period 2005 to 2019 for the baseline as well as all sensitivity tests
are within the CERES uncertainties from Loeb et al. (2021) (Fig. 6b), there are several possible reasons why the posteriori EEI does not reproduce the CERES trend.

From the linearized equation of the EEI, the EEI will increase with time if ERF increases at a greater rate than $\alpha\Delta T$. Over the last decades, there has been rapid reduction of SO$_2$ emissions especially over China (Zheng et al., 2018) contributing to an increase in the total anthropogenic forcing trend (Forster et al., 2023). If the trend in ERF is wrongly implemented, this will
influence our EEI estimate. The idealized aerosol experiments, changing the time evolution of aerosol ERF, strongly influence the trend in EEI (Fig. 6b,c). The posterior EEI trend increases by ~50% in sensitivity tests with weaker aerosol ERF towards 2019 compared to "*Base*" but is still in the lower range of the CERES trend. Sensitivity tests with strengthening of the aerosol ERF all the way to 2019 as well as constant aerosol ERF after year 2000 resulted in a very weak EEI trend from 2005 to 2019, less consistent with CERES data. These results suggest that a weakening of the aerosol ERF
has contributed to the trend in EEI as observed by CERES. Other studies also find a contribution of weakened aerosol ERF to the EEI trend observed by CERES (Raghuraman et al., 2021;Hodnebrog et al., 2024). The trend in EEI from CERES is dominated by increase in absorbed solar radiation with a dominant contribution from clouds (Loeb et al., 2021), hinting to a possible important role of either aerosol-cloud interactions or cloud feedbacks. However, it is difficult to separate the ERFaci from the cloud feedback in the CERES data (Loeb et al., 2021).

As discussed above, the feedback parameter can change over time, but is assumed to be constant in our approach. As the climate feedbacks are dependent on the pattern of the temperature change, this effect must be included in EEI reconstructions to match observed EEI from CERES (Zhou et al., 2021). Andrews et al. (2022) indicated that the pattern effect might have been particularly strong in recent decades and waning post 2014. The possibility that the feedbacks have changed over the





recent decades relative to what is estimated in our approach using observations over a longer period is not included in our

methodology.

The rate of ocean heat gain is a key component for the quantification of the EEI. In our method the information on EEI trend is taken from the OHC data used. Studies investigating the trend in several different OHC dataset find weaker trend than in CERES (Li et al., 2023;Minière et al., 2023) but Minière et al. (2023) highlighted the overlapping uncertainties in both methods for assessing the EEI trends and the challenges in assessing trends over such a short period.

The trend in EEI can be implemented in our estimation method to give an additional constrain on the recent forcing time evolution. The two sensitivity tests with weaker aerosol ERF towards 2019 strongly change the trend in EEI (Fig. 6b) but had limited influence on the posteriori ECS estimate (Table S1). Note that the end year for our estimation is 2022, meaning that the analysis does not include the rapid increase in EEI from CERES data in 2023 (Fig. 6) as well as the record high temperatures in 2023 (Voosen, 2024).

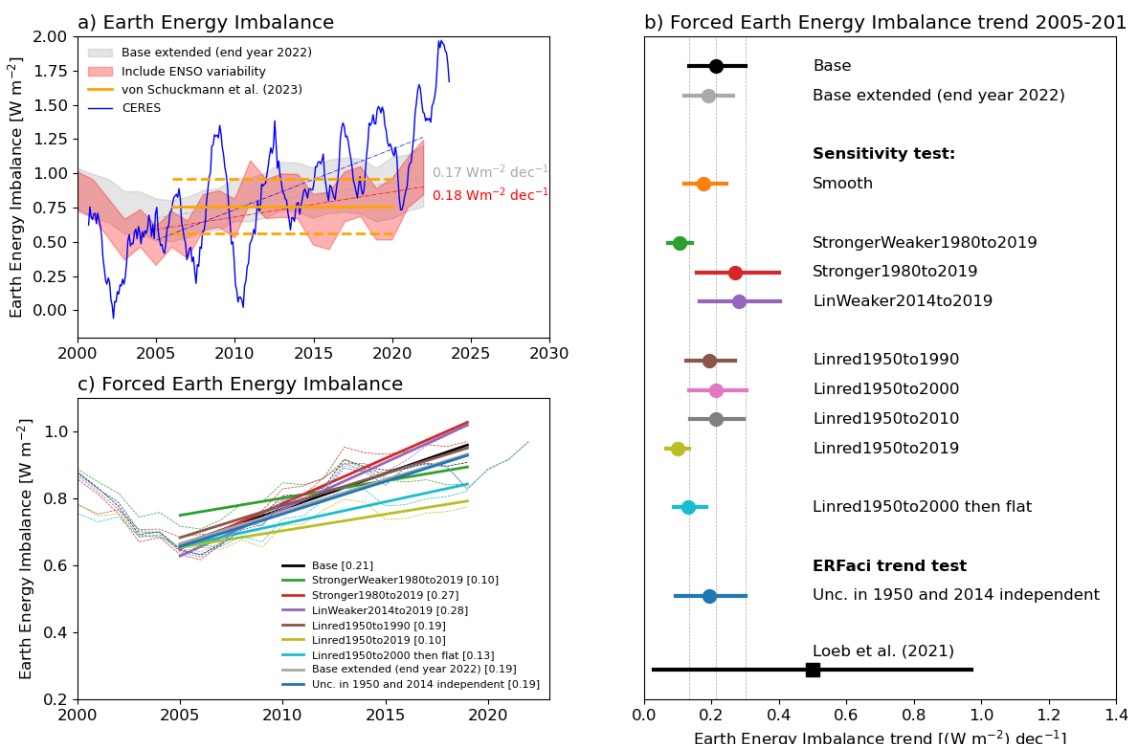


**Figure 6: Posterior Earth Energy Imbalance and Earth Energy Imbalance trend. In a) the gray shading shows the 5 to 95 percentile of the posteriori EEI for the forced temperature response, while the red shading shows the posteriori EEI including ENSO variability for "*Base extended*". The gray and red dashed line indicate the least squares linear fit for the posterior median of EEI without and with ENSO variability included for the period 2005 to 2022 and the trend over this period are indicated in the**

**plot. The orange lines are the EEI from von Schuckmann et al. (2023) of 0.76 W m⁻² (solid line) ± 0.2 W m⁻² (dashed lines) for the 90% confidence interval for the period 2006 to 2020. CERES data are shown as a 12-month running mean in blue (Loeb et al.,**



**2018a) (include data for January 2024) and the least squares linear fit from 2005 to 2022 (dashed blue line). In b) the posteriori forced EEI trend over the period 2005 to 2019 are shown as 5th to 95th percentile as solid line and median value as a dot. At the bottom the mid-2005 to mid-2019 estimates for the trend for the net CERES TOA energy flux of 0.50 ± 0.47 W m⁻² decade⁻¹ (90%**
**confidence interval) from Loeb et al. (2021) are indicated with a black square and the solid line. In c) the posteriori median of the forced EEI from 2000 and onwards are shown for selected sensitivity tests as annual values (dashed lines) and the linear trend from between 2005 to 2019 as solid line. The trends in EEI over this period are indicated in the legend as W m⁻² decade⁻¹.**

## 5 Conclusions

We have used the most up to date ERF time series, observations of temperature change and ocean heat content in a Bayesian
framework to estimate climate sensitivity, aerosol forcing and aerosol forcing pathway. Aerosol ERF is the largest contributor to the uncertainties in the total anthropogenic ERF (Forster et al., 2021;Forster et al., 2024). As prior knowledge of the uncertainties in the time evolution of aerosol ERF is lacking, we use a range of idealized aerosol ERF time series to investigate the sensitivity of our observational based estimates to the assumed aerosol pathway.

Our estimate of climate sensitivity is the inferred climate sensitivity, and it only includes feedbacks that have come into play
over the historical period considered. The historical warming pattern favors lower climate sensitivity values than what is expected from long-term increases in $CO_2$ concentrations from climate model simulations (Andrews et al., 2018). The $ECS_{inf}$ does not include this so called "Pattern effect". If a pattern effect of 0.5 W m⁻² K⁻¹ (the central estimate in IPCC AR6) is added to our "*Base*" estimate, the climate sensitivity estimate is almost identical to the IPCC AR6 very likely range of 2 to 5 K with a best estimate of 3 K (Forster et al., 2021). The pattern effect is limiting historical observations to constrain the
upper end of climate sensitivity (Sherwood et al., 2020;Armour et al., 2024). For near term climate policy that aims for net zero emissions by mid-century (UNFCCC, 2015) the $ECS_{inf}$ might be as relevant as climate sensitivity estimates that consider climate feedbacks over longer timescales. Furthermore, both $CO_2$ forcing (He et al., 2023) and feedbacks might be climate state dependent (Bloch-Johnson et al., 2021) and therefore differ between high forcing scenarios (as 2 or 4xCO2) and near-term net zero emission scenarios.

The $ECS_{inf}$ estimate is dependent on the aerosol pathway. The upper 95th percentile of $ECS_{inf}$ differs by 1.1 K for the different sensitivity tests, and the $ECS_{inf}$ is most sensitive to the aerosol ERF pathway between 1950 and 2014. Allowing ERFaci to evolve more freely within the AR6 uncertainties between 1950 and 2014 and using forcing time series and observational data up to and including 2022, the mean $ECS_{inf}$ is 2.3 K with 90% C.I. from 1.4 to 3.4 K. The estimate is shifted to larger values compared to our estimates where the aerosol pathways are more fixed, with mean $ECS_{inf}$ of 2.1 K
(90% C.I.:1.5 to 2.9) with AR6 ERF prior and observations to 2019 in "*Base*" and 2.2 K (90% C.I.:1.5 to 3.0) with the AR6 extended ERF prior and observations to 2022 in "*Base extended*".

The present-day posteriori distribution of the aerosol ERF is strongly influenced by the different aerosol pathways. As the prior in "*Base*" is tied to the present-day ERF and the observations do not allow for decreasing total anthropogenic ERF in the 1950s to early 1970s, the present-day aerosol ERF shifts to more negative values when the prior for the ERF in the 1960-
70s are weakened in the sensitivity tests. Also, when the historical ERFaci is not tied to the present-day aerosol forcing, the



negative aerosol ERF trend in the 1950s to 1970s is weakened compared to the prior. Strong aerosol forcing in the 1960s-70s is less consistent with observations. Over the more recent period, the pathways with a weakening of the aerosol ERF are more consistent with observations of the Earth's Energy Imbalance from space from CERES since 2005. Currently, observations on OHC constrain the estimated EEI in our method. Future work can implement the trend in EEI from CERES

as an additional constraint in observational based estimation of climate sensitivity, aerosol ERF and aerosol ERF pathways.

For assessing future climate change, evaluating historical climate change is crucial and hence knowledge of how the different drivers have changed over time is critically important. In previous literature, the focus has mostly been on present day aerosol forcing relative to pre-industrial. Better knowledge of the historical aerosol ERF pathway is however needed, not only related to how aerosol ERF has changed over the recent decades (Lund et al., 2023;Quaas et al., 2022), but also further

back in time. For future work, prior uncertainties in the time evolution of aerosol ERF based on expert judgement should be implemented considering uncertainties in historical aerosol and aerosol precursor emissions, as well as uncertainties in the physical processes of aerosol-radiation and aerosol-cloud interactions.

**Appendix A**

In this section, additional information on the Methods is presented.

**Model**

The core of the model framework is a simple climate model (SCM), which is a deterministic energy balance/upwelling-diffusion model (Schlesinger et al., 1992;Schlesinger and Jiang, 1990). The model calculates annual hemispheric near-surface temperature change (blended sea surface temperature and surface air temperature over land) and changes in global ocean heat content (OHC) as a function of ERF time series. The output from the SCM can be written as $\boldsymbol{m}_t(\boldsymbol{x}_{1750:t}, \boldsymbol{\theta})$,

where $\boldsymbol{x}_{1750:t}$ are the ERFs from 1750 until year t and $\boldsymbol{\theta}$ are the true, but unknown, input values to the SCM. $\boldsymbol{\theta}$ is a vector of seven parameters, where one of these is the climate sensitivity parameter ($\lambda$) and the other parameters determine how the heat is mixed into the ocean (e.g. mixed layer depth, air-sea heat exchange coefficient, vertical diffusivity in the ocean and upwelling velocity).

The true state of some central characteristics of the climate system in year t can be written as $\boldsymbol{g}_t = \boldsymbol{m}_t(\boldsymbol{x}_{1750:t}, \boldsymbol{\theta}) + \boldsymbol{n}_t$,

where $\boldsymbol{n}_t$ is a stochastic process, with three terms, representing long-term and short-term internal variability and model error. For the short-term internal variability, the Southern Oscillation index is used to account for the effect of El Niño-Southern Oscillation (ENSO) on the temperature. For the long-term internal variability, the dependence structure is based on a control simulation from a CMIP5 model (Skeie et al., 2014). This term will also represent other slowly varying model errors due to potential limitations of the SCM and the ERF time series. The last error term accounts for more rapidly varying

model errors.



For $\boldsymbol{g}_t$ corresponding long-term observational data are available with individual error terms. Data on surface temperatures are considered separately for the northern and southern hemispheres, and OHC for both 0-700 meters and below 700 meters. For each of the elements of $\boldsymbol{g}_t$ several corresponding observational-based data series are available (Table A1). To gain as much information as possible, we use several data sets for the same physical quantity simultaneously.

We apply a Bayesian approach and use Markov Chain Monte Carlo techniques to sample from the posterior distribution.

In the SCM the climate sensitivity is represented as a climate sensitivity parameter ($\lambda$). $\lambda$ is multiplied by the ERF for a doubling of $CO_2$ (2xCO2) to present the climate sensitivity as $ECS_{inf}$. Here we use the 2xCO2 corresponding to the posterior estimate of $CO_2$ ERF for the conversion from $\lambda$ to $ECS_{inf}$, while in previous work $CO_2$ forcing was included in the combined greenhouse gas forcing time series and the best estimate of 2xCO2 from AR5 (Myhre et al., 2013a) was used (Skeie et al.,

2018). The TCR is calculated using the SCM with the joint posteriori distributions of the parameters forced with an ERF time series of 1% increase per year in $CO_2$ concentration until a doubling of $CO_2$ is reached. The ERF time series used is consistent with the forcing prior used, hence different for AR5 and AR6 priors (Forster et al., 2021). If we use an ERF time series corresponding to the posterior of the $CO_2$ ERF, the posteriori distribution of the TCR is similar to using the best estimate of the $CO_2$ ERF time series (Fig. S13). Also, for $ECS_{inf}$, the probability density function is similar using a fixed

factor and a factor corresponding to the posterior $CO_2$ ERF for the conversion from $\lambda$ to $ECS_{inf}$ (Fig. S13).

**Effective Radiative Forcing**

How the ERF uncertainties from IPCC AR6 and AR6 extended are implemented are described below and summarized in Table A1.

For most of the forcing components, the relative uncertainty is symmetrical and constant in time, and we assume a normal distribution for these components. For the forcing components with a skewed 90% confidence interval presented in AR6, the uncertainty in the forcing is implemented as a combination of two normal distributions as in AR6. The asymmetric uncertainty ranges and the fractional uncertainty were determined by considering ranges below and above the best estimate separately, by dividing the 5th percentile by the best estimate to derive the lower uncertainty range and the 95th percentile by

the best estimate to determine the upper range, treating them as two halves of a Gaussian distribution.

In IPCC AR6 the forcing time series for aerosol–radiation interactions (ERFari) was constructed using a linear relationship between emissions of $SO_2$, BC, OC and $NH_3$ and ERFari for sulphate, black carbon, organic carbon and nitrate aerosols respectively (Smith et al., 2021a). The emission to forcing coefficients were based on multi-model results from Myhre et al. (2013b) and rescaled so the total ERFari of $-0.3 \pm 0.3$ W m$^{-2}$ for present day in the assessment is preserved. The relative

uncertainty of ERFari is not constant in time, but it is symmetrical. To get the same 5 and 95 percentiles as in AR6 for the ERFari timeseries, we assume a normal distribution and correct the uncertainties by a constant for each year, to match the historical uncertainties from AR6.





The forcing timeseries for the aerosol–cloud interactions (ERFaci) in AR6 are based on fits to 11 CMIP6 models with historical time varying ERFaci of a logarithmic function of emissions of $SO_2$, BC and OC (Smith et al., 2021a). A 100 000-
member ensemble was drawn, and the median of this ensemble scaled to the assessed value for present day ERFaci in AR6. The 90% confidence interval for ERFaci are unsymmetrical and not constant in time, and hence the left and the right halves of the prior distribution are modeled by the left and the right halves of two, possibly different, normal distributions. These normal distributions are different from one year to another.

For implementing the solar forcing, a 0.50 fractional uncertainty was applied to the amplitude of the solar cycle, and a linear
1750 to 2019 trend of $\pm0.07$ W m$^{-2}$ (5–95%) range was added to this to represent the uncertainty in the change in the underlying solar forcing of 0.01 W m$^{-2}$, as was done generating the ERF timeseries uncertainties in AR6 (Smith et al., 2021a).

In the SCM, the ERFs are split on hemispheres for ozone, aerosol-radiation interactions, aerosol-cloud interactions and landuse. This split is constant in time, and for ozone calculated from multi-model results of ozone forcing time series from
CMIP6 (Skeie et al., 2020) and land use and aerosols from ERF diagnosed from CMIP6 models (Smith et al., 2020).

**Table A1: Forcing components included in the estimations, their relative uncertainties and description of distribution. The forcing time series, uncertainties and distributions are implemented as in IPCC AR6 (Forster et al., 2021). The uncertainties are presented as percentages: (best-pc05)/best and (best-pc95)/best, where best is the best estimate, pc05 and pc95 is the 5th and 95th percentile**
**respectively. The 90% uncertainty is symmetrical if only one number is presented, and if not otherwise stated, the uncertainties are constant in time. If the uncertainties in AR6 extended (Forster et al., 2023) were updated, this is presented in parenthesis in the table.**

| Forcing components | Uncertainties [%] | Distribution |
|---|---|---|
| co2 | 12% | Normal |
| ch4 | 20% | Normal |
| n2o | 16% | Normal |
| other_wmghg | 19% (19% in 2022) | Normal (Normal, not constant in time) |
| o3 | 50% | Normal |
| h2o_stratospheric | 100% | Normal |
| Contrails | 67% lower, 69% upper (55% lower, 70% upper) | Two normal |
| aerosol-radiation_interactions | 119% in 2019 (99% in 2022) | Normal, not constant in time. |
| aerosol-cloud_interactions | 72% lower, 70% upper in 2019 (74% lower, 70% upper in 2022) | Two normal, not constant in time. |
| bc_on_snow | 100% lower, 125% upper | Two normal |



| land_use | 50% | Normal |
|---|---|---|
| Solar | 50% to the amplitude of the solar cycle + a linear 1750 to 2019 trend of ±0.07 W m-2 (5–95%). | Sum of two independent normal |
| Volcanic | 25% | Normal |

**Observational data**

Table A2 list all the observational based time series used in the estimation. As described in the method section, we only use the temporal profile of the reported error and estimate the magnitude within the model. For OHC we choose to use the temporal development of the uncertainty from Domingues et al. (2008) for all time-series as the reported uncertainties (Fig. S6) may not include the full uncertainties in OHC.

**Table A2: Observation based time series used in the estimation. In parentheses, what is used for the updated time series.**

| Name | Full Name | Period of record | Domain | Reference | DOI/LINK | Date of download |
|---|---|---|---|---|---|---|
| **Temperature:** | | | | | | |
| NOAA | NOAAGlobalTemp v5.0.0 (v5.1.0) | 1880-2019 (1850-2022) | NH, SH | Vose et al. (2021) | https://www.ncei.noaa.gov/products/land-based-station/noaa-global-temp | 23. November 2021 (21. June 2023) |
| HadCRUT5 | HadCRUT.5.0.1 | 1850-2019 (1850-2022) | NH, SH | Morice et al. (2021) | https://www.metoffice.gov.uk/hadobs/hadcrut5/data/current/download.html | 23. November 2021 (21. June 2023) |
| GISTEMP | GISTEMP v4 | 1880-2019 (1880-2022) | NH, SH | (Lenssen et al., 2019;GISTEMP-Team, 2023) | https://data.giss.nasa.gov/gistemp/ | 23. November 2021 (20. June 2023) |
| **OHC** | | | | | | |
| NCEI-Levitus | Levitus/ Lev12-NCEI/ NCEI | 1955-2019 (1955-2022) | OHC0-700m, OHC700-2000m | (Levitus et al., 2012) | https://www.ncei.noaa.gov/data/oceans/woa/DATA_ANALYSIS/3M_HEAT_CONTENT/DATA/b | 23. November 2021 (20. June 2023) |





| | | | | | asin/yearly/h22-w0-700m.dat<br><br>https://www.ncei.noaa.gov/data/oceans/woa/DATA_ANALYSIS/3M_HEAT_CONTENT/DATA/basin/yearly/h22-w0-2000m.dat<br><br>https://www.ncei.noaa.gov/data/oceans/woa/DATA_ANALYSIS/3M_HEAT_CONTENT/DATA/basin/pentad/pent_h22-w0-2000m.dat<br><br>Used the pentadal average up to 2008 and the yearly data thereafter for below 700 meter.<br><br>https://www.ncei.noaa.gov/access/global-ocean-heat-content/ | |
|---|---|---|---|---|---|---|
| PMEL | PMEL/JPL/HIMA R | 1993-2019[#]<br>(1993-2022) | OHC0-700m,<br>OHC700-2000m | (Lyman and Johnson, 2014) | Downloaded from reference in Table 2.SM.1 Data Table in AR6 (Gulev et al., 2021b)<br><br>https://oceans.pmel.noaa.gov/ | 10. December 2021 (11. July 2023, e-mailed from John Lyman) |

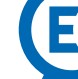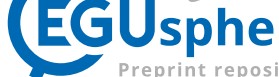

| Domingues | CSIRO/ACE CRC/IMAS-UTAS | 1970-2019 (1970-2022*) | OHC0-700m | (Domingues et al., 2008) | http://www.cmar.csiro.au/sealevel/thermal_expansion_ocean_heat_timeseries.html | 3. May 2022 e-mailed from Domingues |
| JMA-Ishii | MRI/JMA - Ishii | 1955-2019 (1955-2022) | OHC0-700m, OHC700-2000m | (Ishii et al., 2017) | data.jma.go.jp/gmd/kaiyou/data/english/ohc/ohc_global_1955.txt https://www.data.jma.go.jp/gmd/kaiyou/english/ohc/ohc_global_en.html | 17. December 2021 (21. June 2023) |
| IAP-Cheng | IAP/CAS | 1955-2019 (1955-2022) | OHC0-700m, OHC700-2000m | (Cheng et al., 2019a;Cheng et al., 2020;Cheng et al., 2017;Cheng et al., 2019b;Cheng et al., 2022) | ocean.iap.ac.cn/ftp/images_files/IAP_OHC_estimate_update.txt | 10. December 2021 (9. January 2023) |
| Open-OHC | Open-OHC (OPEN-OHCv1.1.2) | | OHC0-700m, OHC700-2000m | (Su et al., 2020) | https://raw.githubusercontent.com/scenty/OPEN-OHC/master/OPENv1.1.1.txt | 10. December 2021 (E-mailed from Wenfang Lu 1. August 2023) |
| EN4 | EN4.2.1/MetOffice/Good et al. (EN.4.2.2.c14/MetOffice) | 1955-2019[#] (1955-2022) | OHC0-700m, OHC700-2000m | (Good et al., 2013;Gouretski and Cheng, 2020;Cheng et al., 2014) | Downloaded from reference in Table 2.SM.1 Data Table in AR6 (Gulev et al., 2021b) https://climate.meto |  4. january 2022 (Data e-mailed from Rachel Killick 25. July 2023 and 1. August 2023) |





| | | | | | ffice.cloud/ocean_h eat.html | |
|---|---|---|---|---|---|---|
| **SOI-index** | | | | | | |
| SOI | Southern Oscillation index, Bureau of Meteorology, Australia | | | | bom.gov.au/climate/ enso/soi_monthly.tx t | 23. November 2021 (23. June 2023) |

[#] Data available until 2018. Extended the data from 2018 to 2019 using the average increase in the other available series from 2018 to 2019.

[*] Data available until 2021. Extended the data from 2021 to 2022 using the average increase in the other available series from 2021 to 2022.

**Estimation**

As a starting point we use the main analysis in Skeie et al. (2018). Stepwise, we updated the observational based time series for surface temperature and ocean heat content, then replace the IPCC AR5 forcing time series by the IPCC AR6 time series (Forster et al., 2021) with end year 2014 before extending the data used in the estimation to year 2019. For a better representation of the distribution of heat in the ocean, the priors for two of the parameters in the SCM are widened compared

to what was used previously (Aldrin et al., 2012), the vertical velocity/upwelling rate and the polar parameter with a new uniform prior of [0.55,7] and [0.161,2] respectively. When replacing the ERF prior with the extended ERF time series (Forster et al., 2023) and add additional years in the observations, a stepwise replacement and extensions of observations and forcings are performed here as well.

Table A3 lists all the different estimations performed in this study.


**Table A3: List of estimations performed with a description of the setup, forcing prior used and end year of observational data used.**

| # | Simulation | Description | ERF prior | End year |
|---|---|---|---|---|
| 1 | Skeie18, AR5 prior | The main analysis in Skeie et al. (2018). | AR5 | 2014 |
| 2 | Update obs. end year 2014 | Same as #1 but use updated observational based data series. | AR5 | 2014 |
| 3 | Replace AR5 prior with AR6 | Same as #2 but replace AR5 ERF priors with AR6 ERF priors. | AR6 | 2014 |
| 4 | End year 2019 | Same as #3 but extend from 2014 to 2019. | AR6 | 2019 |
| 5 | Base | Same as #4 but with new priors for the upwelling velocity [0.55,7] m yr[-1] and the polar parameter [0.161,2]. Both priors are uniform as before (Aldrin et al., 2012). | AR6 | 2019 |
| | | | | |
| | **Sensitivity test:** | For all sensitivity test below, the ERFari and ERFaci from | | |



| | | IPCC AR6 are modified in specific time periods. | | |
|---|---|---|---|---|
| 6 | Smooth | As #5, but ERFari and ERFaci smoothed using locally weigthed scatterplot smoothing in pythons statmodels module over the entire time period. | AR6 | 2019 |
| | | | AR6 | 2019 |
| 7 | Linear1750to1900 | As #5, but a linear ERFari and ERFaci from 1750 to 1900. | AR6 | 2019 |
| 8 | Weaker1900to1950 | As #5, but weaker ERFari and ERFaci between 1900 and 1950 multiplying the AR6 forcings by a sinus curve. | AR6 | 2019 |
| 9 | Stronger1900to1950 | As #5, but stronger ERFari and ERFaci between 1900 and 1950 multiplying the AR6 forcings by a sinus curve. | AR6 | 2019 |
| 10 | Weaker1950to1980 | As #5, but weaker ERFari and ERFaci between 1950 and 1980 multiplying the AR6 forcings by a sinus curve. | AR6 | 2019 |
| 11 | Stronger1950to1980 | As #5, but stronger ERFari and ERFaci between 1950 and 1980 multiplying the AR6 forcings by a sinus curve. | AR6 | 2019 |
| 12 | StrongerWeaker1980to2019 | As #5, but stronger and then wearker ERFari and ERFaci between 1980 and 2019 multiplying the AR6 forcings by a sinus curve. | AR6 | 2019 |
| 13 | Stronger1980to2019 | As #5, but stronger ERFari and ERFaci between 1980 and 2019 multiplying the AR6 forcings by a sinus curve. | AR6 | 2019 |
| 14 | LinWeaker2014to2019 | As #5, but linearly weaker ERFari and ERFaci from 2014 to 2019. The ERFari and ERFaci is weaker in 2019 than in AR6. | AR6 | 2019 |
| 15 | Linear1950to1990 | As #6, but with a linear ERFari and ERFaci between 1950 and 1990. | AR6 | 2019 |
| 16 | Linear1950to2000 | As #6, but with a linear ERFari and ERFaci between 1950 and 2000. | AR6 | 2019 |
| 17 | Linear1950to2010 | As #6, but with a linear ERFari and ERFaci between 1950 and 2010. | AR6 | 2019 |
| 18 | Linear1950to2019 | As #6, but with a linear ERFari and ERFaci between 1950 and 2019. | AR6 | 2019 |
| 19 | Linear1950to1980 then flat | As #6, but with a linear ERFari and ERFaci between 1950 and 1980 and constant ERFari and ERFaci thereafter. | AR6 | 2019 |
| 20 | Linear1950to1990 then flat | As #6, but with a linear ERFari and ERFaci between 1950 and 1990 and constant ERFari and ERFaci thereafter. | AR6 | 2019 |
| 21 | Linear1950to2000 then flat | As #6, but with a linear ERFari and ERFaci between 1950 and 2000 and constant ERFari and ERFaci thereafter. | AR6 | 2019 |
| | **Extension up to 2022:** | | | |
| 22 | Replace AR6 prior with AR6 extended | Same as #5 but replace the AR6 prior ERF priors with the extended AR6 ERF time series (Forster et al., 2023). | AR6 extended | 2019 |
| 23 | Updata obs. end year 2019 | Same as #22 but update all observational based time series. | AR6 extended | 2019 |
| 24 | Base extended (end year 2022) | Same as #23 but extend from 2019 to 2022. | AR6 extended | 2022 |



| | | | | |
|---|---|---|---|---|
| | **ERFaci trend test** | | | |
| 25 | Unc. in 1950 and 2010 independent | Same as #24 but uncertainties in 1950 (and before) and 2014 (and after) independent of each other. Uncertainty scaling factor linearly interpolated for the years in between. | AR6 extended | 2022 |

**Code availability**

The code to reproduce the figures in this manuscript: currently on github  https://github.com/ragnhibs/climsens_24 and will be made available on zenodo upon publication.

**Data availability**

The data needed to reproduce the figures in the manuscript: currently on github  https://github.com/ragnhibs/climsens_24 and will be made available on zenodo upon publication. Observational data used are presented and available via links

provided in Table A2. ERF timeseries are available from AR6 (Smith C.J. et al., 2021) and AR6 extended (Smith C. and Forster, 2022).

**Author contribution**

RBS wrote the paper and made the figures. RBS prepared the data and designed the experiments. MH and RBH performed the Bayesian estimation. All co-authors discussed the design, results and contributed to the writing.

**Competing interests**

No conflicting of interests.

**Acknowledgements**

The work was funded through the Norwegian Research Council project (grant number 314997) and by the European Union's Horizon 2020 research and innovation program under grant agreement No 821205 (FORCeS). We kindly acknowledge all

the observational data providers listed in Table A2.



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
