# Peer review of "The aerosol pathway is crucial for observationally constraining climate sensitivity and anthropogenic forcing"

_EGUsphere, 2024_

## Author Comment (AC1)

We would like to thank both reviewers for their useful comments on this manuscript. The reviewers' original comments are in blue, our response in black and text added/modified in the manuscript in *italic*.

**Reviewer #1**

This paper is interesting and useful for comparing different aerosol modeling frameworks, especially for analyzing the impact of the structure of the time history of aerosol forcing. The authors replace aerosol ERF from AR6 with various idealized alternative pathways and also extend temperature, ocean heat content, and ERF time series using the most up-to-date versions of these data. I have a few major and minor comments.

**Major comments**

Idealized pathways: I wonder if the authors could more clearly describe the advantages and disadvantages of these various idealized pathways, vs. using a simple aerosol forcing model whose free parameters can vary in the Bayesian inversion and therefore yield different time histories of aerosol forcing. I am thinking of the simple aerosol forcing model in Stevens et al., 2015, and employed in Smith et al., 2021b and Albright et al., 2021, which are cited in this paper. How does the present approach change the results and allow for more flexibility? What would the results look like if using a simple aerosol forcing model (using SO2, and/or also including BC and OC)?

The aim of this study was to investigate the sensitivity of our results to the aerosol ERF pathway. When changing the pathway in different time periods, changing the time of the maximum strength of the aerosol ERF, we see how these different idealized aerosol pathways influence the results. The last sensitivity test does allow for more flexibility compared to the Base setup and the other sensitivity tests performed in this study. The next step, as we wrote, is to include a prior for the aerosol ERF pathway that is based on expert knowledge. This must include uncertainties in emissions (both in magnitude and time development), uncertainties in ERF (including cloud adjustments) and as the geographical distribution of the aerosol and aerosol precursor emissions has changed over time, this also influences the forcing strength evolution. These points are outlined in the introduction. The simple aerosol models used in these papers are flexible, but they are still tied to the assumed emission pathway (linearly or logarithmic). For a next step a comparison to those models should be included.

In the discussion we have specified that the ERFaci evolves more freely compared to the base setup:

*"If we then let the ERFaci more freely evolve within the AR6 uncertainty range between 1950 and 2014 compared to "Based extended", the trend in aerosol ERF is similar to Smith et al. (2021) with +0.024 W m$^{-2}$ decade$^{-1}$ between 1980 and 2014."*

Co-variance among parameters: One of the advantages of the Bayesian framework is that it yields a joint distribution of uncertain parameters. Aerosol forcing and ECS_inf are presented independently. Could these values be shown in a joint pdf?

[Figure]

Figure 4: The joint posterior distribution of aerosol ERF in 2022 and posterior ECSinf. In a) for "Base extended" and in b) where the aerosol ERFaci uncertainties in 1950 and 2014 are treated independently. The 5[th] and 95[th] percentile in the aerosol ERF prior for 2022 are shown as horizontal dashed lines.

To better illustrate the aerosol ERF and ECSinf dependencies, we have added a new figure (Figure 4) showing the joint posteriori distribution for *"Base extended"* and the estimation where ERFaci evolves more freely between 1950 and 2014. Both have the same aerosol ERF prior distribution at the end year (2022), and this figure clearly shows the importance of the aerosol pathway, as in b) a more pronounced banana-shape is shown than in a).

We have added the figure and the following text to the manuscript in section 3.3 Sensitivity test adjusting ERFaci trend:

*"In Fig. 4 the distribution of the ECSinf and aerosol ERF in 2022 for this sensitivity test and "Base extended" are shown. The joint distribution is stretched towards higher values of ECSinf in this test (Fig. 4b) compared to "Base extended" (Fig. 4a)."*

Other interesting parameters regarding how heat is mixed into the ocean ("e.g. mixed layer depth, air-sea heat exchange coefficient, vertical diffusivity in the ocean and upwelling velocity", line 517) are presented. I was interested in how these parameters

traded off in the various scenarios, and whether strongly differing scenarios of idealized pathways showed different parameter covariance? Could that parameter covariance provide physical insights?

We have looked at the parameter estimates and the covariance for the different estimations with idealized aerosol pathways. This did not provide any further insights and therefore we have not included it in the manuscript. The main reason for the difference between the sensitivity test, that we focus on in the paper, are that we change the pathway of the aerosol forcing. With weaker aerosol ERF in the 1970-1990s, stronger present day aerosol ERF can be consistent with the observations used to constrain the forcing and model parameters.

NH vs. SH temperatures: It could be useful, I think, to provide physical insights why modifying the aerosol pathway over certain time periods changes the ECS_inf and other parameters more than changing it over other periods. For example: "The observed temperature and OHC used in the estimation do not allow for a weakening in the anthropogenic ERF in the second half of the century." Is this result dependent on using hemispheric temperatures, or are similar results obtained when using global temperatures? Is there a particular decade in the second half of the century that emerges as most important for not allowing for more negative aerosol radiative forcing? Is it before / during / after peak emissions of SO2?

For the method we use we can not easily run with global instead of hemispheric temperatures. However, OHC is an important constraint (see response below) and the OHC is a global quantity and not split in hemispheric values.

For Figure 2 and 3 we have changed the values on the x-axis and now we start at 1850 instead of 1750, then it is easier to see the time evolution of the prior and posterior ERFs. According to the CEDS emission inventory, the peak of $SO_2$ emissions is around 1980, but relatively flat between 1970s and 1990s. The emissions rapidly increased after 1950.

In section 3.1, we have tried to make the link between the Fig. 2 and Fig. 3 even clearer. We wrote that "*for "Base", the prior shows a reduction in anthropogenic ERF from the 1950s until the 1970s, while in the posterior the ERF is flat over this period. The observed temperature and OHC used in the estimation do not allow for a weakening in the anthropogenic ERF in the second half of the century.*"

We have added the link between anthropogenic ERF and aerosol ERF: "*... anthropogenic ERF in the second half of the century, and hence do not allow for a stronger aerosol ERF than the prior median (Fig. 3c).*"

We have also added a link between the AR6 aerosol pathway and the $SO_2$ emissions in this paragraph: "*In AR6 the aerosol ERF shows a steep strengthening from 1950s to 1970s, the period when the global anthropogenic $SO_2$ emissions rapidly increased (Hoesly et al., 2018), then the aerosol ERF…*"

In the conclusion section we have added the following:

"*Stronger aerosol forcing than around -1.4 W m$^{-2}$ in the 1960s and 70s, the period leading up to the peak $SO_2$ emissions (Hoesly et al., 2018), is less consistent with observations.*"

Also, related to Figure 5, we see and we also wrote, that the trend in the period 1950 to 1969 and 1960s to 1979 is weaker in the posterior compared to the prior for this sensitivity test. This is hence the period before the peak of the SO2 emissions.

We have added text to the following section:

"*For the earliest two periods, 1950-1969 and 1960-1979 the posterior trends are weaker than in the prior. This is the period where the global anthropogenic $SO_2$ emissions rapidly increased before peak emissions around 1980 (Hoesly et al., 2018).  For the latest period, 2009-2019, the posterior aerosol ERF trend is also weaker than in the prior and…*"

Role of ocean heat uptake observations in the model: I was also interested to know how much ocean heat content constrained the model and added additional information, compared to surface temperatures, but, if I am understanding correctly, I did not see this discussed in much detail in the paper. Could the authors comment on it, or refer to a previous discussion of the role of surface temperatures vs. ocean heat content in a previous work by this team? Thank you!

The ocean is the dominant energy storage in the climate system. Johansson et al 2015 clearly showed that the use of OHC data narrowed the range of climate sensitivity inferred from historical observations. Also, in Skeie et al 2014, excluding OHC data between 2000 to 2010, the ECS estimate widened, and were similar to using data (OHC and temperature) up to year 2000 only.

We have added the following sentence in section 2.3:

"*It is important to attempt to represent the full uncertainty as the OHC data are previously shown to have profound influence on observationally constrained climate sensitivity estimates (Johansson et al., 2015;Skeie et al., 2014).*"

**Minor comments**

Line 78, for known reasonS. Small typo.

Corrected.

In Figs. 2 and 3, panel c, I found the different lines / cases difficult to distinguish.

We have tried to improve the readability of the figures by excluding the ERF time evolution between 1750 and 1850 and changing the green color that was used.  It should be easier to distinguish AR5 prior median and posterior median now.

Fig. 2a. Why is the posterior Skeie2018 so close / strongly constrained by the AR5 prior, whereas Base is less constrained by the AR6 prior? I see a description around line 175 but no reason given (is it elsewhere?)

The reason is the different time evolution of the two forcing priors. We described it around line 175, but we have tried to make it clearer and rewritten the following section:

*"The prior ERF distribution in 2014 is similar for AR5 and AR6 (Fig. 2a), while the time evolution of the prior is quite different (Fig. 2c). The posteriori distribution of the anthropogenic ERF for each step updating ERF prior and extending the data (Table A1) are shown in Table S3. From this stepwise update and extension of the data used in the estimation, the temporal evolution of the forcing pathway, when replacing AR5 with AR6 forcing prior, seems to play a large role in explaining why the prior and posterior distribution of the anthropogenic ERF for the end year are so different using AR6 forcing prior and similar using AR5 forcing prior (Fig. 2b)."*

Line 496-7, could "strong" be more clearly defined with a number value/range? Around -2 W/m2?

The 5th percentile for the posteriori distribution is around -1.4 W m$^{-2}$.

We have added:

*"Stronger aerosol forcing in the 1960s-70s than around -1.4 W m$^{-2}$..."*

**Reviewer #2**

The paper investigates the important ways aerosols impact the constraining of climate sensitivity. The sensitivity tests and calculations are a welcome addition to the literature. I only have a few minor comments listed below that need to be addressed prior to publication.

Minor Comments

  1. Grammar: I think the title should be "observationally constraining"?

We have changed the title according to your suggestion.

2. General: Need to leave space between paragraphs, it was hard to read blocks of text

We agree. We used the Copernicus Word template, but in the revision, we have added space between paragraphs.

3. Line 41: Earth Energy Balance -> Earth energy balance

Corrected.

4. Line 106: Why not use the Oceanic Niño Index/Niño3.4, the most widely used index of ENSO?

Other indexes used for ENSO are based on Sea Surface Temperature (SST) data. The SST data are included in the data for surface temperature change. Therefore, we chose to use an index for ENSO that was not based on temperature in our Bayesian approach when we developed the method. SOI are derived from pressure measurements.

5. Figures 2-3: Please provide more frequent x-axis labels, perhaps every 10 years. It is difficult to judge the years discussed in the text with 50 years spacing on the graph.

By excluding the years prior to 1850 in Fig2c and 3c, we now have a 20 year spacing between the labels on the x-axis, and it should be easier to read the figure.

6. Line 250: Please elaborate on what exactly saturation means here as it's not fully clear from the sentences that follow. Saturation to a reader may sound like a huge perturbation like a 10xBC type experiment.

In the context here, the saturation effect is that additional aerosol perturbation will only cause minimal increase in the forcing. The forcing does not linearly scale with the aerosol perturbation.

We have modified the text: "*These aerosol ERF time series may represent a saturation of the aerosol ERF after the strongest ERF is reached, where an additional increase or decrease in aerosol or aerosol precursor emissions has only minimal influence on the aerosol ERF.*"

7. Line 282: Perhaps worth noting that even multi-millennial simulations don't equilibrate sometimes, i.e., $T_s$ still keeps increasing.

We have added *at least* to this sentence.

"*To equilibrate the models requires to run the model for at least thousands of years*"

8. Figure 4 and surrounding discussion on pattern effect: Perhaps Dessler (2020) (https://doi.org/10.1175/JCLI-D-19-0476.1) should be mentioned in this discussion as it relates quite closely to the discussion here. Furthermore, how should one reconcile these results with his results where \Delta \lambda = 0.2 Wm^-2K^-1 on average?

Yes, this study is relevant. The value of 0.2 W m$^{-2}$ K$^{-1}$ is within the IPCC AR6 range of 0.0 to 1.0 W m$^{-2}$ K$^{-1}$. We have added the following in the discussion:

"*From climate model simulations over the historical period, the ECS inferred from different climate realizations due to internal natural variability can differ by 0.7 K (Dessler, 2020). This further highlight challenges in inferring climate sensitivity from historical observations, as we only have one realization of the Earths historical climate.*"

9. Line 343: At line 68 it's mentioned the lifetime of aerosols is days yet here it's mentioned that these aerosols can have an impact for decades. Could you please reconcile this for the reader?

We have added the following: «*due to the lag in the oceanic thermal response of the aerosol forcing*»

10. Line 352: Sherwood et al. (2020) (https://doi.org/10.1029/2019RG000678) showed that paleoclimate has a strong constraint. Could that be relevant here? If so, please discuss it.

Yes, we have added the following:

"Combining multiple lines of evidence, including also paleoclimate data, may give a stronger constraint on the ECS (Sherwood et al., 2020)."

11. Line 439: Perhaps worth mentioning updated literature here: Raghuraman et al. (2023) (https://doi.org/10.1175/JCLI-D-22-0555.1) has separated this. They find that the observed SW trend is due to 40% ERF, 30% SW cloud feedback, and 30% surface albedo + SW water vapor feedbacks.

Raghuraman et al. (2023) used CERES data in combination with model data to do this separation. Using observations alone, the separation is difficult as also they pointed out. We have added the reference and pointed out that it is difficult to separate forcing and feedbacks from observations alone.

Rewritten as:

"However, from the CERES data alone it is difficult to separate the ERFaci from the cloud feedback (Loeb et al., 2021;Raghuraman et al., 2023)."

The $\boldsymbol{m}_t$ is the output from the SCM, as described in the previous sentence. We have added:

"*The output from the SCM, the timeseries of temperature and OHC, can be written as* $\boldsymbol{m}_t(\boldsymbol{x}_{1750:t}, \boldsymbol{\theta})$*, where…*"

We have also added a reference to a newly published description paper of the SCM: (Sandstad et al., 2024)

12. Line 516: Why is climate sensitivity an input? Shouldn't it be an output? Please clarify/explain further.

The climate sensitivity is a parameter in the SCM, given a prior distribution and the results is the posterior distribution.

We have rewritten:

"*… where* $\boldsymbol{x}_{1750:t}$ *are the ERFs from 1750 until year t which are the true, but unknown, input values to the SCM. The true but unknown parameters of the SCM is* $\boldsymbol{\theta}$ *which is a vector of seven parameters, where one of these is the climate sensitivity parameter ($\lambda$). The other parameters…* "

We have also added a sentence clarifying that the ERFs and model parameters are given prior distributions:

"*The model parameters ($\boldsymbol{\theta}$) and the ERF time series ($\boldsymbol{x}_{1750:t}$) are given prior distributions, and we apply a Bayesian approach and use Markov Chain Monte Carlo techniques to sample from the posterior distribution.*"

Dessler, A. E.: Potential Problems Measuring Climate Sensitivity from the Historical Record, J. Clim., 33,2237-2248, https://doi.org/10.1175/JCLI-D-19-0476.1, 2020.

Hoesly, R. M., Smith, S. J., Feng, L., Klimont, Z., Janssens-Maenhout, G., Pitkanen, T., Seibert, J. J., Vu, L., Andres, R. J., Bolt, R. M., Bond, T. C., Dawidowski, L., Kholod, N., Kurokawa, J. I., Li, M., Liu, L., Lu, Z., Moura, M. C. P., O'Rourke, P. R., and Zhang, Q.: Historical (1750–2014) anthropogenic emissions of reactive gases and aerosols from the Community Emissions Data System (CEDS), Geosci. Model Dev., 11,369-408, 10.5194/gmd-11-369-2018, 2018.

Loeb, N. G., Johnson, G. C., Thorsen, T. J., Lyman, J. M., Rose, F. G., and Kato, S.: Satellite and Ocean Data Reveal Marked Increase in Earth's Heating Rate, Geophys. Res. Lett., 48,e2021GL093047, https://doi.org/10.1029/2021GL093047, 2021.

Raghuraman, S. P., Paynter, D., Menzel, R., and Ramaswamy, V.: Forcing, Cloud Feedbacks, Cloud Masking, and Internal Variability in the Cloud Radiative Effect

Satellite Record, J. Clim., 36,4151-4167, https://doi.org/10.1175/JCLI-D-22-0555.1, 2023.

Sandstad, M., Aamaas, B., Johansen, A. N., Lund, M. T., Peters, G. P., Samset, B. H., Sanderson, B. M., and Skeie, R. B.: CICERO Simple Climate Model (CICERO-SCM v1.1.1) – an improved simple climate model with a parameter calibration tool, Geosci. Model Dev., 17,6589-6625, 10.5194/gmd-17-6589-2024, 2024.

Sherwood, S., Webb, M. J., Annan, J. D., Armour, K. C., Forster, P. M., Hargreaves, J. C., Hegerl, G., Klein, S. A., Marvel, K. D., Rohling, E. J., Watanabe, M., Andrews, T., Braconnot, P., Bretherton, C. S., Foster, G. L., Hausfather, Z., Heydt, A. S. v. d., Knutti, R., Mauritsen, T., Norris, J. R., Proistosescu, C., Rugenstein, M., Schmidt, G. A., Tokarska, K. B., and Zelinka, M. D.: An assessment of Earth's climate sensitivity using multiple lines of evidence, Rev. Geophys., n/a,e2019RG000678, 10.1029/2019RG000678, 2020.

Smith, C. J., Harris, G. R., Palmer, M. D., Bellouin, N., Collins, W., Myhre, G., Schulz, M., Golaz, J. C., Ringer, M., Storelvmo, T., and Forster, P. M.: Energy Budget Constraints on the Time History of Aerosol Forcing and Climate Sensitivity, J. Geophys. Res., n/a,e2020JD033622, https://doi.org/10.1029/2020JD033622, 2021.